# Developing the Common Marmoset as a Translational Geroscience Model to Study the Microbiome and Healthy Aging

**DOI:** 10.3390/microorganisms12050852

**Published:** 2024-04-25

**Authors:** Kelly R. Reveles, Alexana J. Hickmott, Kelsey A. Strey, Aaryn C. Mustoe, Juan Pablo Arroyo, Michael L. Power, Benjamin J. Ridenhour, Katherine R. Amato, Corinna N. Ross

**Affiliations:** 1College of Pharmacy, The University of Texas at Austin, Austin, TX 78712, USA; streyk@uthscsa.edu; 2Graduate School of Biomedical Sciences, University of Texas Health San Antonio, San Antonio, TX 78229, USA; ahickmott@txbiomed.org (A.J.H.); amustoe@txbiomed.org (A.C.M.); jarroyo@txbiomed.org (J.P.A.); cross@txbiomed.org (C.N.R.); 3Southwest National Primate Research Center, Texas Biomedical Research Institute, San Antonio, TX 78227, USA; 4Center for Species Survival, Smithsonian’s National Zoo and Conservation Biology Institute, Washington, DC 20008, USA; powerm@si.edu; 5Department of Mathematics and Statistical Science, University of Idaho, Moscow, ID 83844, USA; bridenhour@uidaho.edu; 6Department of Anthropology, Northwestern University, Evanston, IL 60208, USA; katherine.amato@northwestern.edu

**Keywords:** common marmoset, aging, geroscience, microbiome, fecal microbiota transplantation

## Abstract

Emerging data support associations between the depletion of the healthy gut microbiome and aging-related physiological decline and disease. In humans, fecal microbiota transplantation (FMT) has been used successfully to restore gut microbiome structure and function and to treat *C. difficile* infections, but its application to healthy aging has been scarcely investigated. The marmoset is an excellent model for evaluating microbiome-mediated changes with age and interventional treatments due to their relatively shorter lifespan and many social, behavioral, and physiological functions that mimic human aging. Prior work indicates that FMT is safe in marmosets and may successfully mediate gut microbiome function and host health. This narrative review (1) provides an overview of the rationale for FMT to support healthy aging using the marmoset as a translational geroscience model, (2) summarizes the prior use of FMT in marmosets, (3) outlines a protocol synthesized from prior literature for studying FMT in aging marmosets, and (4) describes limitations, knowledge gaps, and future research needs in this field.

## 1. Introduction

Human health and longevity are determined by complex interactions between genetic, epigenetic, and environmental factors; however, emerging evidence supports another biological mechanism of aging—the microbiome. Advances in genomic sequencing techniques have revealed complex interrelationships between the host, gut ecosystem, and human exposures. The gut microbiome promotes health via energy and nutrient extraction, the synthesis of key vitamins and hormones, host immune system modulation, the metabolism and elimination of toxins, and protection against pathogens [1]. Thus, disease may occur when there is a substantive change in the microbiome community structure that affects the ecosystem’s function (i.e., dysbiosis), particularly when the ecosystem is unable to return to a healthy state after perturbation. Poor functional changes include a reduction in important microbial metabolic byproducts, such as short-chain fatty acids, vitamins and minerals, and energy-storing glycogen [2,3] and a reduction in pathogen protection via the production of antimicrobial molecules by commensal microbes [1]. Dysbiosis has been associated with a wide range of health conditions, including metabolic diseases (e.g., obesity, diabetes), inflammatory bowel disease, neuropsychiatric conditions, allergies, cancer, alcoholic liver disease, hepatic encephalopathy, and rheumatoid arthritis [4]. Importantly, many of these conditions are more prevalent among older adults; thus, age is a potential confounding factor in understanding the relationship between the microbiome and health conditions.

Age-related changes in the gut microbiome may influence the development of aging-related diseases. Microbiome structure and functional capacity develop rapidly after birth and then remain relatively stable from toddlerhood to adulthood [5]. Thereafter, prior literature has shown that older adults often experience a loss of microbial diversity, an abundance of core microbial taxa, and an increase in subdominant and potentially pathogenic taxa (e.g., Proteobacteria). It is important to note that aging-related microbiome changes may be mediated by health conditions, healthcare exposures (e.g., medications, hospitalizations), lifestyle, behavioral, dietary, and environmental changes (e.g., nursing home residence); thus, evaluating associations between the microbiome and aging is extremely complex. The microbiome is also constantly exposed to environmental challenges that can result in dysbiosis. If the microbial community can recover from insult, it is considered resilient. Resilience is distinct from cross-sectional community structure and function. A healthy, resilient microbiome protects us from disease, whereas a non-resilient microbiome can result in a chronic dysbiotic state that can lead to disease [6,7].

Most microbiome studies have focused on chronological age, which may not accurately represent a person’s overall health status. More recent studies have found that biological aging (i.e., decline in physical and cognitive function) may be more strongly associated with reduced health span than chronological age. The term frailty has been used to describe a state of age-associated decline in function across multiple physiological systems that can result in decreased resilience and greater vulnerability to physiological stress and poor health outcomes. Frailty is often operationalized by describing accumulated deficits over time (e.g., exhaustion, low physical activity, disability, falls) [8,9]. Prior studies indicate that frailty may better estimate risk for adverse health outcomes compared to chronological age, and frailty has been associated with changes in microbiome alpha and beta diversity, age-associated microbial subcommunities, and distinct biochemical functions [10,11]. While beyond the scope of the current review, prior literature has well-described the relationship between the microbiome and frailty-associated biological and physiological outcomes [12].

Due to the connection between dysbiosis and disease, fecal microbiota transplantation (FMT) may be a possible therapeutic intervention for positively altering microbial communities to improve health in aging individuals. For example, FMT has been used successfully for decades to treat human *Clostridioides difficile* infection, a common and debilitating gastrointestinal infection that disproportionally affects older adults [13]. This is due to the strong link between dysbiosis and *C. difficile* infection pathogenesis. A rich and diverse microbiome provides for colonization resistance, whereas dysbiosis creates a microenvironment that favors *C. difficile* colonization and vegetative, toxin-producing growth [14]. Administration of fecal material from healthy donors via the upper (e.g., oral, endoscopy) or lower (e.g., enema, colonoscopy) gastrointestinal route to sick recipients restores microbial diversity and creates a microenvironment that resists development of future clinical infections [14]. In the clinical management space, FMT has become more accessible in recent years. Several stool banks are available in the United States that provide rigorously screened donor material to health care providers. In the past year, the first microbiome restoration therapies, Rebyota^®^ (fecal microbiota, live-jslm) and Vowst^®^ (fecal microbiota spores, live-brpk) were approved by the US Food and Drug Administration as live biotherapeutic products for the prevention of recurrent *C. difficile* infection, providing more standardized donor material and improving provider access.

Outside of human infectious diseases, mouse models of FMT offer the best evidence of a causal relationship between microbial communities and other health conditions. Evaluations of gnotobiotic mice recipients of human donor fecal material as well as mouse-derived fecal material result in weight gain and metabolic dysfunction if they receive material from an obese donor [15,16]. Conversely, transplanting fecal microbiota from lean individuals to obese individuals results in weight loss and rescued metabolic function [17,18]. Further, transplanting fecal microbiota from mice that were fed high fat diets resulted in increased anxiety, increased stereotypical behavior, and decreased cognition associated with increased neuroinflammation in animals that did not yet show changes in obesity or metabolic function [19]. These findings in mice suggest that shifts in microbial communities associated with FMT, and the byproducts that they produce, are associated with inflammation and many disease states. They also suggest that FMT may be an interesting interventional treatment to stabilize microbial communities and restore homoeostasis for individuals exhibiting dysbiosis. While FMT evidence from mice is promising, direct translation to humans is limited; therefore, we propose to use a nonhuman primate model, the common marmoset (*Callithrix jacchus*), as a translational geroscience model system to test questions related to FMT and health span.

The marmoset is an excellent model for evaluating microbiome-mediated changes with age and interventional treatments. Marmosets have a much shorter lifespan and more rapid development than other nonhuman primate species studied in biomedical research. Juvenile marmosets reach sexual maturity at 12–18 months and full adult body length and weight at 24 months. Most marmoset colonies report a median life span between 5 and 6 years, with a maximum life span of around 16 years, when mortality associated with experimental protocols are excluded [20]. The ratio of the rates of adult aging between human years and marmoset years would, thus, be 1:8 to middle age and 1:6 to maximum age (compared with 1:2 and 1:2.5 for macaques). Marmosets are typically considered geriatric at 8 years of age, when the first dysfunction associated with aging is detected (cognitive deficit, brain amyloid deposit, and blood chemistry alterations) [21]. In addition to their relatively short lifespan, they are small and relatively cheap to house, making them easy and safe to handle for experimentation [22,23]. As a nonhuman primate model, there are many social, physiological, and behavioral functions that more closely translate to human function than do rodent models, including diurnal activity patterns, feeding and nutritional patterns, and the formation of pair-bonded social groups. Marmosets have been found to display many aging phenotypes that mimic human aging, including increased risks of cardiovascular changes, inflammatory disease, metabolic impairment, suppressed immune function, frailty, and impaired cognition [20,21].

Given these findings, our group has begun to use the marmoset as a translational geroscience model to study the aging microbiome and the impact of FMT on health span. Investigators have integrated microbiome methods and outcomes into studies involving marmosets as translational models for human health over the last several years. These initial investigations into microbial diversity present in marmoset populations and the influence of environmental factors on the microbiome provide data for baseline prevalence, sensitivity, and recovery. The remainder of this review summarizes the prior use of FMT in marmosets as well as synthesizes prior research protocols to inform one for studying FMT in aging marmosets.

The first report of FMT used in marmosets in the literature is from a 2017 case by Yamazaki et al. [24]. A male common marmoset suffered chronic and recurrent diarrhea due to *C. difficile* infection despite appropriate antibiotic treatment with metronidazole. The seven healthy marmosets chosen as donors had no history of prior medication for at least 3 months. Stool was collected within 30 min after defecation from at least two selected donors per each day of collection. The team processed donor stool by mixing 3 g of feces, 4 g of powdered marmoset food, 2 g of honey, and 8 mL of lukewarm water to make a wet mash. The FMT was then orally administered to the sick animal for four consecutive days. This report noted the rapid resolution of diarrhea after one day, with a corresponding negative *C. difficile* stool test by day four.

Our team published the only other report of FMT in marmosets. In this prior work, the marmoset microbiome was compared by age group, then it was assessed whether healthy donor fecal material could be safely transplanted into other healthy marmoset recipients as preliminary data to support future health span studies. This was a cross-sectional study comparing the gut microbiome composition of 10 male young adult (2–5 years) and 10 male geriatric (8+ years) marmosets housed at the Barshop Institute for Longevity and Aging Studies in San Antonio, Texas [25]. The microbiome from stool samples was characterized using 16S rRNA V4 sequencing. Geriatric marmosets had a lower mean Shannon diversity compared to young marmosets (3.15 vs. 3.46; *p* = 0.019). Geriatric marmosets also had a lower abundance of Firmicutes (0.15 vs. 0.19; *p* = 0.003) but higher abundance of Proteobacteria (0.22 vs. 0.09; *p* = 0.023) compared to younger adults. The increased abundance of Proteobacteria and loss of Firmicutes mimics what has been reported in aging humans and supports the possibility that these microbial shifts reflect an aging process rather than an environmental modification. The next study was a prospective study of healthy young adult males (age 2–5 years) with no recent medication use [26]. Stool from two donors was combined with sterile saline and administered via unsedated gavage in 0.5 mL doses to five young recipients once weekly for 3 weeks. Safety outcomes and alterations in the gut microbiome composition measured via 16S rRNA sequencing were compared at baseline and monthly up to 6 months post-FMT. Despite the small sample size, we did note significant differences in the percent relative abundance of certain bacterial taxa at the phylum and family level in FMT recipients from baseline to 1 month and baseline to 6 months post-FMT. The most important information gained from these prior studies indicates that FMT recipients did not experience any negative health outcomes over the course of the treatment, providing early signals that FMT is safe and possibly effective.

## 2. Marmoset FMT Protocol

There have been no previously published, standardized protocols for preparing FMT for administration to marmosets; therefore, our team conducted a review of human, non-human primate, and rodent protocols to inform best practices for preparing and administering FMT to marmosets (Table 1). Oral administration was chosen for feasibility and animal safety; no studies have evaluated enema FMT procedures in primates. While there are limited data, based on our clinical and scientific expertise, we believe this protocol may be used for microbiome restoration after antibiotic treatment as well as for the treatment of dysbiosis-related disease such as aging or *C. difficile* infection. An overview of the protocol and the rationale for decisions is provided here.

### 2.1. Donor Screening

The marmoset donor screening protocol is similar to the protocols recommended for humans (Table 2) [43]. Animals with any health conditions, medications, stool pathogens, or abnormal systemic lab values should be excluded from donation. Additionally, health span measures were included, such that only donors with a healthy phenotype were eligible for donation. If microbiome sequencing is available, baseline microbiome measures may also be assessed. Alterations in this protocol for scientific studies may be appropriate (i.e., the use of scientific positive or negative control donors or recipients).

While there is no consensus on what constitutes a healthy microbial community, especially in marmosets, several markers generally associated with health can be evaluated. For example, in combination with health span indicators, ideal donors would have higher microbiome diversity and microbiome community structure with a higher abundance of traditionally “healthy” microbiota and a lower abundance of pathobionts. Specifically, certain bacterial taxa have been generally associated with health including those involved in short-chain fatty acid production (e.g., *Faecalibacterium*, *Eubacterium*, *Clostridium* cluster XIV, and *Roseburia*), while others have been negatively associated with health, such as lipopolysaccharide producers and organisms with pathogenic potential (pathobionts; e.g., Enterobacterales) [44,45]. Because bacterial abundance and phylogeny are the strongest predictors of FMT engraftment [46], optimal donors will have a proportionately higher abundance of beneficial microbes and lower abundance of harmful bacteria. There is some evidence from human FMT treatment for *C. difficile* that FMT success is affected not only by donor microbiome diversity but also microbiome composition, specifically the ability of the donor to provide the necessary taxa to restore metabolic deficits in the recipient. Other genetic and environmental factors may also play a role, leading to the hypothesis that super donors may exist [47]. Currently, there is no way to predict the clinical efficacy of a donor before FMT. It has been suggested that pooling donor stool together will limit the chances a recipient will receive ineffective stool [48]. This was tested in a human randomized controlled trial of inflammatory bowel diseases patients. Clinical remission rates were similar to previous reports, but the pooled stool resulted in higher microbial diversity than individual donor samples, driven primarily by one super-donor [49].

### 2.2. FMT Material Processing

An overview of FMT processing can be seen in Table 3. Stool samples collected from animals should be mixed with glycerol to a concentration of approximately 10% and stored at –80 °C as soon as possible to prevent the loss of viable oxygen-sensitive species. The short- and long-term storage of microbiota require the use of a cryoprotectant to mitigate cellular damage due to freeze-hydration and ice crystal formation. Glycerol protects cells from both forms of damage and achieves a glass-forming tendency when used at concentrations of 5–20%. Human FMT guidelines recommend that glycerol be used at a concentration of 10% [43].

The amount of donor material will need to be calculated based on the number of recipients, though the amount of fecal material used varies from study to study (Table 1). For human FMT, it is recommended that 50 g of feces (~1 g/kg) be diluted in 3–5 times the amount of sterile saline solution (150 mL) [43]. Marmosets at our colony weigh on average 450–475 g (~0.45 kg) but can range from 300–600 g. Prior studies on rats have used approximately 0.5 mL (2 mL/g) of FMT material administered once per week for 3 weeks. This was also demonstrated as feasible and safe in our preliminary studies. Thus, enough stool should be collected to administer 0.5 mL doses using a 2:1 sterile PBS to fecal material ratio. Researchers should also consider collecting overfill for the metagenomic sequencing of donor samples.

It is recommended to freeze samples if not processing them immediately. Prior studies have found that there is no significant difference in microbial diversity and abundance, including of oxygen-sensitive bacteria, when stored at room temperature for no longer than 24 h compared to immediately freezing [50]; however, the viability of oxygen-sensitive organisms may be affected by various techniques and the duration of oxygen exposure. A study by Bellali et al. [51] found that cultured bacteria reached a 50% yield when the samples were exposed to oxygen for 120 min without any protectant medium, while the percentage of culturability increased to 67% in the presence of antioxidants. More importantly, when samples were exposed to oxygen for less than 2 min, combined with the work under the anaerobic chamber, the culturability increased to 87%. Another study by Ott et al. [52] found that bacterial viability significantly declines at room temperature after 4 h or refrigerator after 8 h.

Both anaerobically and aerobically prepared samples are efficient in the treatment of recurrent *C. difficile* infection, but some studies for indications other than *C. difficile* infections demonstrated higher resolution rates with anaerobically prepared fecal suspensions [53,54]. Most bacteria that colonize the human gut are anaerobic; they are 100–1000 times more numerous than aerobic bacteria [55]. A considerable part of the bacterial genera of healthy microbiota produces resilient spores, allowing for the interindividual transfer of at least a proportion of oxygen-sensitive intestinal bacteria. Given that these spore-forming bacteria typically represent about one-third of gut bacteria, and that disorders associated with microbiota alteration are typically defined by a lower abundance of anaerobic bacteria, it is rational to expect that the anaerobic processing of samples would be relevant for FMT success in the treatment of these disorders. Thus, human FMT guidelines recommend anaerobic processing if possible. For our prior studies, we prepared samples in an anaerobic chamber, homogenized feces with sterile anaerobic PBS using a sterilized blender, and transferred to a centrifuge tube and capped tightly. Tubes were removed from the chamber and centrifuged at 800× *g* for 2 min. The tubes were then placed back into the anaerobic chamber and supernatant was aliquoted into 0.5 mL doses (as many as possible).

If possible, aliquots should be immediately administered to recipients to avoid another freeze–thaw cycle; however, if using aliquots for research on multiple animals, this may not always be feasible. If freezing is needed, the final suspension should be stored at −80 °C. Frozen aliquots should be used within 6 months of freezing [56]. It is unclear if long-term storage will affect the viability and clinical effectiveness of the FMT material. A study by Burz et al. [57] found that there was only a 9.8% loss of viable microbes after 3 months of storage at −80 °C. Another study found virtually no difference in the effectiveness of FMT material stored for <6 months at −80 °C (83.8%, *n* = 1473) compared to material stored for 6–12 months at −80 °C (83.8%, *n* = 439) and for >12 months at −80 °C (83.3%, *n* = 12), suggesting that frozen storage duration does not significantly impact the rate of clinical cure [58].

### 2.3. FMT Recipient Preparation and Delivery

FMT may be conducted without antibiotic preconditioning, though preconditioning may enhance the transfer of the donor’s microbiome into the recipient’s gastrointestinal tract. Prior studies have demonstrated that the depletion of the recipient’s gut microbiome with antibiotics may result in a better engraftment of the donor’s microbiome into the recipient [59,60,61]. Most FMT literature describes the efficacy of FMT in humans with *C. difficile* infection. These patients have significantly reduced microbial diversity and exposure to antibiotic treatment prior to FMT. For non-infectious indications, most prior animal studies have used relatively broad-spectrum antibiotics with or without a bowel prep regimen, though there is no standardized regimen. Additionally, there are no prior data evaluating antibiotic preconditioning in FMT engraftment in nonhuman primates such as marmosets. While we noted some shift in microbiome composition with FMT in our preliminary studies that were not preconditioned with antibiotics, planned future studies will precondition marmosets with enrofloxacin (standard weight-based dosing) prior to FMT for three consecutive days with two days of recovery prior to FMT delivery.

On the day of FMT, the fecal suspension should be thawed in a warm (37 °C) water bath. After thawing, saline solution could be added to obtain a desired suspension volume as needed. The thawed fecal material should be delivered to the animal within 6 h after defrosting. Recipients can receive FMT once a week for three weeks via an oral gavage of 0.5 mL of material. If a control group is used, controls can receive aliquots of sterile saline via oral gavage over three weeks to control for handling effects and gavage treatment. This dosing regimen was selected based on successful transplant in rat models of obesity [17].

### 2.4. Linking FMT to Health Span

FMT is expected to produce alterations in microbiome diversity and function. Health measures in marmosets have been evaluated for years and have been found to reflect alterations in aging animals. If microbial diversity and microbiome function impact overall aging health, it is possible these impacts could be detected in a longitudinal study of gut functionality, nutritional intake, metabolism, cognition, immune function, physiological homeostasis, and mobility. Measuring how the health span measures change after FMT and assessing time-linked microbiome diversity will allow us to establish the efficacy of FMT to preserve or restore healthy aging in marmosets. Additionally, measures of microbiome resilience (i.e., ability to restore equilibrium after perturbation) and function (e.g., metabolomics) will provide additional rigorous data for testing microbiome–host–disease associations.

## 3. Knowledge Gaps and Challenges with Using FMT in Marmosets

While marmosets are an excellent model for studying the aging gut microbiome and more specifically FMT, there are limited studies that have examined the relationship between the microbiome and health span, how FMT impacts long-term microbiome functioning, or FMT utilization in nonhuman primates, resulting in several knowledge gaps and challenges.

First, the selection of FMT donors and FMT dosing has not been rigorously evaluated in any species, including humans. The protocol presented herein was based on previously published human and rodent protocols. Oral gavage dosing was chosen primarily for feasibility and animal safety. Oral is also the primary delivery route used in rodent FMT models. It is possible that oral delivery could result in a loss of donor microbes as they transit the more acidic upper gastrointestinal tract. In humans, oral delivery is performed using large, enteric-coated capsules that are swallowed, which is infeasible for animals. Additional studies of lower gastrointestinal FMT delivery (e.g., enema) on microbiome structure, function, safety, and efficacy are needed, as lower delivery is common in humans. Next, additional health characteristics may need to be screened in marmosets that are not typically present or of concern in humans. Further, while most human FMT studies use one or two FMT doses, our protocol was based on rodent studies using three doses over three weeks. While this approach has resulted in the engraftment of donor microbes, it is unclear whether this regimen will sustain engraftment over the long-term. Recent human *C. difficile* studies have demonstrated the sustained engraftment of donor microbes following FMT for up to two years of follow-up [62], but this may not directly translate to FMTs for non-*C. difficile* indications or non-human models. Future studies should evaluate whether and when booster doses are needed to sustain engraftment, and more importantly, influence health measures.

As noted in the protocol, it is unclear if antibiotic pretreatment is necessary if using FMT for research or non-infectious indications. Prior studies suggest that antibiotic pretreatment may result in more effective microbiome engraftment in the recipient, though the data are mixed [59,60,61]. It is also unclear which antibiotics would be most effective, though given the diverse composition of the gut microbiota, broad-spectrum antibiotics or antibiotic combinations that deplete aerobes and anaerobes and Gram-positive and Gram-negative bacteria may be necessary. Future studies should compare the degree to which core microbial taxa and subcommunities engraft in marmosets with and without antibiotic pretreatment and using different antimicrobial regimens. Further, studies should determine the effect of antibiotic pretreatment on health measures.

Marmosets are social animals, like humans, and are housed in captivity as pairs or family groups. Environmental exposures and social interactions may impact microbial diversity. To further add to this, marmosets may consume their partner’s fecal excrement either directly or through anal genital grooming, which may impact the microbiome and complicate the FMT outcomes. To control for this potential complication, for our planned study, we enrolled geriatric marmosets as vasectomized male–female pairs to be FMT recipients or controls. The pairs were matched for treatment (either both received FMT or both received placebo) to limit the effect of coprophagia on study outcomes. Future studies should determine the impact of co-housing on microbial structure, function, and health measures.

Lastly, one of the strengths of using animal models to study the microbiome is control over the environment and other exposures (e.g., diet, medications). However, limited studies have evaluated the impact of specific diets, supplements, and medications on the gut microbiome of nonhuman primates; thus, it is difficult to establish firm recommended exclusionary criteria or factors that may confound health measures that should be controlled for in the design or analysis phase. Specifically, antibiotic use post-FMT may deplete the donor microbiota, but the extent of this effect and how it impacts clinical outcomes has not been studied.

## 4. Conclusions

The gut microbiome plays a critical role in human health and likely mediates health span and longevity, though data are still limited in this field. The marmoset is an excellent model to study the relationship between the microbiome, aging, and the development of disease. Recent preliminary studies suggest that FMT may be safely performed in marmosets, though standardized protocols for FMT processing and administration are limited. The marmoset FMT protocol described herein was developed based on a critical review of human, rodent, and primate FMT literature. Given the many complex steps for acquiring material, processing, and administering FMT, the protocol necessitates validation for scientific and therapeutic purposes. Additionally, many knowledge gaps still exist on best practices to optimize FMT and microbiome studies in marmosets. Future studies should aim to determine (1) optimal donor health and microbiome characteristics to influence health outcomes, (2) standardized and reproducible FMT processing procedures to optimize engraftment, and (3) the dosing regimen necessary to sustain long-term engraftment. The incorporation of microbiome function through metabolomics, for example, is also recommended to provide more robust information on environmental characteristics and associations with health beyond simply microbial community structure.

## Figures and Tables

**Table 1 microorganisms-12-00852-t001:** Overview of prior animal FMT study protocols.

Study	Species	Fecal Material Amount	Diluent Amount	Administration Amount	Antibiotics	Bowel Prep
Bornbusch 2021 [27]	Ring-tailed lemurs	2–3 mL (2–3 fecal pellets)	3 mL feces/8 mL sterile saline	5–8 mL/dose	AMO × 7 days	No
Di Luccia 2015 [17]	Rats (460 g)	2 pellets/rat	2 mL/g (PBS * + cysteine)	500 µL/dose	AMP + NEO × 8 weeks	No
Lleal 2019 [28]	Rats (200 g) & Mice (25 g)	100 mg	2 mL PBS *	2 mL/dose	No	OME * + Citrafleet
Schmidt 2020 [29]	Rats (200 g)	NR	1:10 dilution with PBS *, L-cysteine, glycerol, & water	500 μL/dose	No	No
Ubeda 2013 [30]	Mice	1 pellet	1 mL PBS *	200 µL/dose	AMO * × 7 days	No
Le Bastard 2018 [31]	Mice	NR	5 g/mL total	200 µL/dose	AMP * × 7 days	No
Wei 2018 [32]	Mice	NR	400 mg/mL NS *	100 g/50 kg	No	No
Wrzosek 2018 [33]	Mice	NR	Diluted 100-fold in BHI & skim milk	200 µL/dose	No	PEG
Stebegg 2019 [34]	Mice	Pellets from 8–14 donors	1 mL PBS * per 300 mg of feces	150 μL once	No	No
Badran 2020 [35]	Mice	Fecal pellets of 5 mice	1 mL PBS * per 100 mg of feces	100 μL/dose	No	No
D’Amato 2020 [36]	Mice	50 g	500 mL (saline + 12.5% glycerol)	1 mL/dose	AMPHO B * + METRO * + AMP * + VANC * + NEO * over 24 days	No
Li 2020 [37]	Mice	12 fecal pellets per cage containing 3 mice each	3 mL sterile PBS	200 μL/dose	No	No
Turnbaugh 2009 [38]	Germ-free mice	1 g	10 mL	200 µL/dose	No	No
Ussar 2015 [16]	Germ-free mice	NR	NR	200 µL/dose	No	No
Wong 2017 [39]	Germ-free mice	1 g	5 mL PBS	200 µL/dose	No	No
Ridaura 2013 [40]	Germ-free mice	500 mg	5 mL PBS *	200 µL/dose	No	No
Diao 2016 [41]	Germ-free mice	Not reported	1:9 *w*/*v* PBS *	50 µL/dose (oral) + 2 mL on fur	No	No
Zeng 2013 [42]	Germ-free mice	2 g	10 mL PBS	50 µL/dose (oral) + 2 mL on fur	No	No

* Abbreviations: AMO, amoxicillin; AMP, ampicillin; AMPHO B, amphotericin B; BHI, brain heart infusion; METRO, metronidazole; NEO, neomycin; NR, not reported; NS, normal saline; OME, omeprazole; PEG, polyethylene glycol; PBS, phosphate-buffered saline; VANC, vancomycin.

**Table 2 microorganisms-12-00852-t002:** Recommended FMT donor screening [43].

Screening	Donor Selection
Medical History History or exposure to transmissible diseases; Known systemic infection at the time of donation; Recent (<3 months) gastrointestinal infection; History of gastrointestinal disorders or diarrhea; History of immunological conditions.	Animals with any of these conditions should be excluded from donation
Recent (<3 Months) Medication History Antibiotics or probiotics; Immunosuppressants or investigational drugs.	Animals with any of these conditions should be excluded from donation
Stool Testing Clostridioides difficile and Clostridium perfringens; Salmonella, Shigella, Campylobacter, E. coli 0157; Giardia lamblia; Cryptosporidium parvum; Klebsiella spp.; Pseudomonas spp.; Protozoa.	Animals with any of these pathogens detected in the stool should be excluded from donation
Blood Testing Complete blood count and blood chemistry; Inflammatory cytokines; Immune markers.	Animals with abnormal values in any of these blood tests should be excluded from donation
Health Span Indicators Metabolic function; Gut functionality; Body composition; Oral glucose tolerance; Blood pressure; Cognition; Mobility.	Donors should be healthy; exclude any animals with abnormal health span indicators
Microbiome Community Diversity and Structure Shannon diversity; Abundance of butyrate-producing bacteria; Abundance of other “healthy” microbiota; Abundance of pathobionts.	In combination with health span indicators above, recommended donors will have high microbiome diversity and abundance of traditionally “healthy” microbiota and low abundance of pathobionts

**Table 3 microorganisms-12-00852-t003:** Overview of FMT processing procedure.

Steps
Collect stool from individual animals and store at –80 °C in tubes mixed with 10% glycerol; In an anaerobic chamber, homogenize feces with sterile PBS using a blender and transfer to a centrifuge tube and cap tightly; Remove tube from chamber and centrifuge at 800× *g* for 2 min; Place tube back in the chamber and aliquot supernatant into 0.5 mL doses; Administer immediately if possible; If unable to administer immediately, store aliquots at –80 °C; on the day of FMT, thaw suspension in a warm (37 °C) water bath and administer within 6 h.

## Data Availability

No new data were created or analyzed in this study. Data sharing is not applicable to this article.

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
