# Peer review of "Developing the Common Marmoset as a Translational Geroscience Model to Study the Microbiome and Healthy Aging"

_microorganisms, 2024, doi:10.3390/microorganisms12050852_

Round 1

Reviewer 1 Report

Comments and Suggestions for Authors

I was given a manuscript to check titled: Developing Captive Marmosets as a Translational Geroscience Model to Study the Microbiome and Healthy Aging.

From my point of view, this paper contains some methodological gap that affect the quality of the article against a subject of the research exceedingly actual and important.

Therefore, I indicate some recommendations for improvements in the work presented.

Suggested improvements are:

Abstract. The authors should start with a short intro that better highlights their work and end up with a paragraph that include results.

The discussion could benefit from further information regarding the existing theory. In addition, the discussion does not address the limitations and strength of the study and it is necessary that it start with a first paragraph describing the main aims and then the main results.

Several aspects of FMT are not discussed in a proper way. The text provides detailed methods of how and what was mixed for FMT, but the Review should contain generalizations explaining the purpose of these details. Conclusions should be drawn at the end of each subsection based on the information provided.

Authors should add a paragraph where they state the most important outcome of their work.

References are appropriate but they should add the reference for the data performed in Table 1.

Authors should check the sequence of the number of the tables.

Sincerely

Reviewer 2 Report

Comments and Suggestions for Authors

The title of the review does not correspond to its content. The review contains information that, from my point of view, is not very well structured, and in principle is far from the “developing captivity of the monkey". The authors have mixed everything up. The text focuses too much on Clostridioides difficile infection, which is not correlated with age. This is an infection, as well as Salmonella, Shigella, Klebsiella! The appearance of cells in the intestine does not correlate with age. So, it is necessary to deeply separate the sources of dysbiosis appearance in old humans.

Several aspects of FMT are discussed in a row in the text: sources of stool, methods of sample preparation, methods of administration, methods of storage, etc. Everything in a row, without accents and conclusions in the text as I should be in the Review. The text provides detailed methods of how and what was mixed for FMT, but the Review should contain generalizations explaining the purpose of these details. Moreover, it follows from the text that it is extremely difficult to cultivate and store cells for FMT, which, of course, will lead to negative or unpredictable results. Conclusions should be drawn at the end of each subsection based on the real (positive and negative) information provided.

I recommend to introduce the Table 1A from the Appendix A as Table 2 to the main text of the paper (Line 192). In this case, the numeration of references in this Table should be corrected.

The change in concentrations of hormones regulating the presence of glycogen in the intestine walls is one of the main problems of dysbiosis with increased age. The probiotics, used the receptors associated with glycogen, lost their chance to be well attached to the intestine walls. After that, it is possible to introduced the probiotics by different ways (with or without antibiotics, orally or via gavage, etc.), the result will not be long and efficient. This information should be disclosed in the review, because it is important.

Please add the reference for the data performed in Table 1.

Line 232: This is Table 2, but not Table 1. The appearance of the compiled information in the Table 2 should be explained and the references for the information should be mentioned. In case the development of these steps was made by the authors, the paper converts to the experimental paper instead of Review! In addition, there are many places in the article where the authors describe their experiments (line 103 - we propose to use a nonhuman primate model, line 132 - we developed for studying FMT, line 148-149 - We first conducted a cross-sectional study, line 152 - we found, line 158-159 - we conducted a prospective study, line 164 - we did note significant differences, lines 172-173 - We designed the protocol, line 174 - We also chose to use oral administration, line 176 - we believe this protocol may be used, lines 183-184 - we included health span measures, line 193 - We designed the protocol to specifically study, line 194 - We also chose to use oral administration, lines 240-241 - we chose to collect enough stool, line 242 - we needed a total of 78 doses, line 243 - we needed a minimum of 19.5g stool, line 244 - we collected a total of 50g stool, line 256 - we recommend freezing samples, line 269 - e prepared samples in an anaerobic chamber, line 294 - we noted some shift in microbiome composition, lines 355-356 - we enrolled geriatric marmosets), which classifies this article as experimental! And in this case, the material of the article should be presented in accordance with another template (with materials and methods). Thus, as a Review such paper should be rejected.

Please, see the Line 206 (Faecalibacterium, Eubacterium, Clostridium), lines 154,156 (Firmicutes), lines 48, 155,156 (Proteobacteria), lines 178,198,212,259 (C. difficile), line 207 (Roseburia), line 209 (Enterobacteriaceae), etc. All names of the microorganisms should be italicized. I recommend to check this carefully.

Finally. The authors analyze mice as a model on which an oral method of fecal administration to sick animals was performed. Do the authors of this article really suggest using the same FMT method of stool administration (ingestion of human feces as part of food) for humans after all?

Round 2

Reviewer 2 Report

Comments and Suggestions for Authors

I still believe that this article is not a Review, but a scientific development (Article), in which, as the authors themselves write, both in the Introduction and in the Conclusion, the protocol they developed is presented based on the literature they read and their own experiments. (Please see lines 473-474: “The marmoset FMT protocol described herein was developed based on critical review of human, rodent, and primate FMT literature”).

The purpose of the article is a presentation of theoretical conclusions and recommendations obtained by the authors. The article should be reclassified from a review to an experimental article.

In addition, in terms of the number of tables and figures that should usually be presented in the Review (and there are only 2 tables here, since the table presented in the Appendix is essentially part of Supplementary materials), this article does not meet the general requirements for Reviews published in Journals of  MDPI. My suggestion to the authors to include a table from Supplementary materials in the main part of the article was ignored by them.

Part 2 with the title “Literary review” should be a part of the Introduction, since the "literary review" inside the article itself, which is considered a review, looks very funny. I suggested this to the authors, but they ignored this recommendation of mine.

The entire third part looks like the authors' Results, and Part 4 is presented as a Discussion. This is exactly how the authors should name their parts, reclassifying the article into a research one.

The new part 2 (the methodological part )of the article should reflect the criteria for comparing different protocols, on the basis of which the authors compare them.

Author Response

Please see our responses attached. 
